# Bone Loss in Patients with Pancreatic Neuroendocrine Tumors

**DOI:** 10.3390/jcm11226701

**Published:** 2022-11-12

**Authors:** He Tong, Miaomiao Wang, Jingjing Liu, Chuangen Guo, Zhongqiu Wang, Jianhua Wang, Xiao Chen

**Affiliations:** 1Department of Imaging, The Second Affiliated Hospital of Bengbu Medical College, Bengbu 233040, China; 2Department of Radiology, The Second Affiliated Hospital of Soochow University, Suzhou 215025, China; 3Department of Radiology, Affiliated Hospital of Nanjing University of Chinese Medicine, Nanjing 210029, China; 4Department of Radiology, The First Affiliated Hospital, School of Medicine Zhejiang University, 79 Qingchun Road, Hangzhou 310003, China

**Keywords:** bone, osteoporosis, pancreatic neuroendocrine tumors

## Abstract

Background: Pancreatic diseases may affect nutritional status, which is one of the important associated factors of bone health. High prevalence of osteoporosis or osteopenia has been reported in patients with pancreatitis. The bone loss in pancreatic neuroendocrine tumors (PNETs) has not been reported. In this study, we showed the prevalence of bone loss and possible associated factors in PNET patients. Methods: A total of 91 PNET patients were included. Bone status was evaluated based on computed tomography (CT) attenuation (Housfield units, HU): >160 HU, normal bone mineral density; osteopenia, 135 HU ≤ CT value ≤ 160 HU; osteoporosis, <135 HU. Associated factors for bone loss were identified by logistic regression analyses. Results: The average age was 55.76 years old in PNET patients. The prevalence of osteoporosis and low bone mass was 37.4% and 60.4%, respectively. Higher prevalence of osteoporosis was observed in patients older than 50 years (64.0%). Multivariate logistic analysis showed that age was an associated factor for low bone mass (odds ratio (OR) = 1.13, 95% confidence interval (CI): 1.04–1.22) and osteoporosis (OR = 1.14, 95% CI: 1.03–1.20). Diabetes was also associated with bone loss in PNET patients after adjusting with confounders (OR = 13.56, 95% CI: 1.02–132.4). Conclusions: Our data show that bone loss is common in patients with PNETs. Age and diabetes are associated with bone loss in PNET patients.

## 1. Introduction

Osteoporosis is one of the major public health problems and is associated with bone fragility and high fracture risk [1]. It has been reported that patients with gastrointestinal disease (GI) had a high risk of low bone mass or osteoporosis [2]. High prevalence of osteoporosis or osteopenia has been reported in patients with chronic pancreatitis [3,4,5] and acute pancreatitis [6]. The association between gastric cancer and osteoporosis was also reported [7,8]. Considering neuroendocrine tumors (NET) may have hormone hypersecretion, osteoporosis/osteopenia is also reported in patients with NETs [9]. Pancreatic neuroendocrine tumors (PNETs) represent a rare subgroup of neuroendocrine tumors (NETs) [10]. The estimated annual prevalence of PNETs was 0.48 per year 100,000 persons. Given the improvement in diagnostic techniques, the occurrence of PNETs is increasing [11]. However, the prevalence of bone loss in PNETs has not been clarified, except for a few case reports [12].

Pancreatic diseases may affect nutritional status and is one important associated factor of bone health. PNET may metastasize in bone, increasing the risk of bone loss, break or fracture [13]. In addition, diabetes and obesity are potential risk factors for gastroenteropancreatic NET occurrence [14] and they may also have an impact on bone loss or fracture [15,16,17]. Moreover, diabetes is associated to a more advanced and progressive disease [14,18]. However, the role of diabetes in PNET-associated bone loss is still unclear. In the present study, we performed a retrospective analysis to show bone loss in patients with PNETs and possible associated factors.

## 2. Material and Methods

### 2.1. Patients

During January 2016 and January 2022, a total of 116 cases of PNETs were found in our institution. Those patients with distal metastasis, without computed tomography (CT) examinations, and who received any treatment, such as surgical resections, radiotherapy, chemotherapy or somatostatin before CT examination were excluded from this study. This retrospective study finally included 91 patients with PNETs. The information of age, gender, history of diabetes mellitus (DM), tumor grade, and tumor size were collected from the medical database. DM was also evaluated by the plasma glucose levels on two separate occasions. Functional PNETs were evaluated by clinical symptoms, such as glucopenia and refractoriness anabrosis. This study was approved by the Ethics Board of the Affiliated Hospital of Nanjing University of Chinese Medicine. Informed consent was obtained from each subjects. This study was performed in accordance with the Declaration of Helsinki.

### 2.2. PNETs Grade

PNETs grade were divided into three groups based on the ki67 index and mitosis count [19]. Briefly, Grade 1 (G1): Ki-67 ≤ 2 and/or mitosis count < 2/10 HPF; G2: Ki-67 index is 3–20 and/or mitosis count is 2–20/10 HPF; G3: Ki-67 index > 20% and/or mitosis count >20 per 10 HPF. G3 tumor was not divided into well-differentiated G3 and pancreatic neuroendocrine carcinoma (PNEC) because of some missing histological information.

### 2.3. CT Scanning

CT scans were obtained from the multi-detector CT system (GE healthcare, Tokyo, Japan; Philips Brilliance 64, The Netherlands). The CT scanning protocols were as follows: Tube voltage of 120 kV; slice thickness of 2–5 mm and automatic tube current modulation. The images were reconstructed in the workstation using a 0.625 mm section thickness and 0.5 mm increments. The average CT attenuation of the lumbar vertebra (L1-L3) in a region of interest (ROI) through trabecular bone were recorded in Housfield units (HU) for each scan, thus avoiding erosion and sclerosis. The determination of bone attenuation in L1-L3 are shown in Figure 1. Bone loss and osteoporosis were defined based on CT attenuation. Briefly, according to previous defined attenuation thresholds, the cohort was divided into normal bone mineral density (bone attenuation > 160 HU), osteopenia (135 HU ≤ bone attenuation ≤ 160 HU) and osteoporosis (bone attenuation < 135 HU) [20].

### 2.4. Statistical Analysis

The data analyses were performed by SPSS 20.0 (IBM Corp., Armonk, NY, USA). The continuous data was shown as mean ± standard deviation and qualitative data was shown as a number. Correlation analysis was used to show the association between variables. Univariate and multivariate logistic regression analyses were used to identify the associated factor with bone loss. Statistical significance was defined if *p* value was less than 0.05.

## 3. Results

### 3.1. Characteristics of Patients

The characteristics of patients in PNETs are summarized in Table 1. Among 91 patients with PNETs, the mean age was 55.76 years old. There were 35 women and 56 men, respectively. Most of the PNET patients were asymptomatic. Eight patients had functional tumors. The mean CT values of L1-L3 was 158.0 HU. The tumor location was distributed roughly equally among the head-neck (*n* = 49, 53.8%) and body-tail (*n* = 42, 46.2%) and the average tumor size was 3.14 cm.

### 3.2. The Prevalence of Bone Loss

The prevalence of osteoporosis and bone loss was 37.4% and 60.4% (Table 1), respectively. The prevalence of osteoporosis and low bone mass or osteoporosis in patients with PNETs in men and women are shown in Figure 2. Women tended to have a higher risk of osteoporosis than men (45.7% vs. 32.1%) (Figure 2). Figure 3 showed the prevalence of osteoporosis and low bone mass or osteoporosis in patients older than 50 years. Women tended to have a higher risk of osteoporosis than men (64.0% vs. 36.8%), but no significant difference were observed.

### 3.3. Risk Factors for Bone Loss

CT attenuation was negatively correlated with age (r = −0.62, *p* < 0.01) and DM (r = −0.33, *p* = 0.01). However, such an association was not observed between bone CT attenuation and tumor size (r = −0.05, *p* = 0.69) or tumor grade (r = −0.04, *p* = 0.74). Subsequently, logistic regression analysis was used to identify the associated factors. Risk factors for osteoporosis and low bone mass in patients with PNETs are shown in Table 2. The univariate and multivariate logistic regression analyses both showed that age (odds ratio (OR) = 1.11, 95% confidence interval (CI): 1.05–1.17; OR = 1.14, 95% CI: 1.03–1.20) and DM (OR = 10.64, 95% CI: 1.32–118.7; OR = 13.56, 95% CI: 1.02–132.4) were independent risk factors for osteoporosis in patients with PNETs. Univariate and multivariate logistic regression analysis showed that age (OR = 1.12, 95% CI: 1.06–1.18; OR = 1.13, 95% CI: 1.04–1.22) was an independent risk factor for low bone mass.

## 4. Discussion

Patients with gastroenteropancreatic-NETs (GEP-NETs) have an increased risk of developing osteopenia and osteoporosis. The bone health of patients with GEP-NETs can also be influenced by hormone hypersecretion, specific microRNAs, nutritional status, or vitamin D deficiency besides the direct effect of bone metastasis [21]. However, the incidence and risk factors of the bone loss in patients with PNETs have not been well recognized. Our data showed that bone loss is common in patients with PNETs (60.4%). Considering that patients with PNETs usually live for a long time, bone health should attract the attention of those patients.

The prevalence of osteoporosis and low bone mass based on quantitative CT (QCT) have been reported in a recent study [22]. For those patients ≥ 50 years, the prevalence of osteoporosis was 28.9% for women, and the prevalence of low bone mass was 42.98% for men and 41.07% for women. Our study showed that the prevalence of osteoporosis and low bone mass in PNETs patients was both higher than the national data from China [22]. The association between DM and PNETs occurrence has been reported [14,23]. Some studies also showed that the occurrence of distant metastasis was higher in PNET patients with DM than those without diabetes [24]. We also showed that DM was a risk factor for osteoporosis or low bone mass in PNETs patients. The association between DM and bone loss or fractures have been widely studied. However, such association was rarely reported in PNET patients. Tumor grades are associated with PNETs treatment and prognosis. However, we did not observe an association between tumor grades and bone loss. One possible explanation is that metastatic PNETs were excluded from our study. Age is the main determined factor for bone loss in the general population. Similar results were observed in our population with PNET, thus suggesting that the role of gonadal status played a critical role in bone health.

Partial or total resections of the pancreas is also associated with osteoporosis [25]. The patients in our study did not receive any treatment, such as surgical resections, radiotherapy, chemotherapy or somatostatin before CT examination. Therefore, the effects of partial or total resections of the pancreas on bone was not studied in our study. We speculated that surgical resection may cause more severe bone loss in PNET patients. The bone health in patients with pancreatic neoplasms is a matter worthy of attention. A healthy lifestyle, such as physical activity, strength training, training to prevent falls, smoking cessation and decreasing alcohol consumption, are important for bone health. Additionally, the American Society of Clinical Oncology (ASCO) suggested that cancer patients should receive a personalize bone mineral density test or take pharmacologic interventions if necessary [26].

The mechanism of how PNET affects bone metabolism are not well clarified. Several related factors have been reported. Briefly, pancreatic diseases may affect the patients’ nutritional status because pancreatic juice is important for digestion. Hormone hypersecretion may be another important factor for bone loss in PNET [21]. Functional PNETs may affect bone metabolism by secreting hormone, such as serotonin (5-HT). NET may also affect the bone by secreting microRNA [20], such as miRNA-201 and miRNA196a.

There are several limitations. First, the sample size was small because of the rarity of PNETs. Our study is a just exploration, and further studies are needed. Second, body mass index may be an associated factor for bone mineral density (BMD). However, we did not collect the data of height and weight. Interestingly, a study showed that QCT-based BMD was not associated with body mass index (BMI) [27]. Therefore, the missing data of BMI may not affect our conclusion. Third, we only showed the prevalence of bone loss, and we did not investigate the possible mechanisms. Fourth, the prevalence of osteoporosis after pancreas resection was not followed up. It would be important to know this data for patient management. Fifth, it would be better to evaluate bone mass by Dual energy X-ray absorption (DXA) or QCT. However, QCT is not routinely performed during abdominal CT scans. Bone CT attenuation can also be used to define bone mass [28]. Finally, we did not have the data of hypogonadism in men, and bone turnover markers, vitamin D status, medical therapies, and other risk factors, such as smoking or corticosteroid use. Prospective studies on this topic are needed.

## 5. Conclusions

In conclusion, our study showed that the prevalence of osteoporosis and bone loss in patients with PNETs was high. Age and diabetes are the two associated factors with bone loss in PNET patients. Bone health needed attention considering that these patients usually live for a long time.

## Figures and Tables

**Figure 1 jcm-11-06701-f001:**
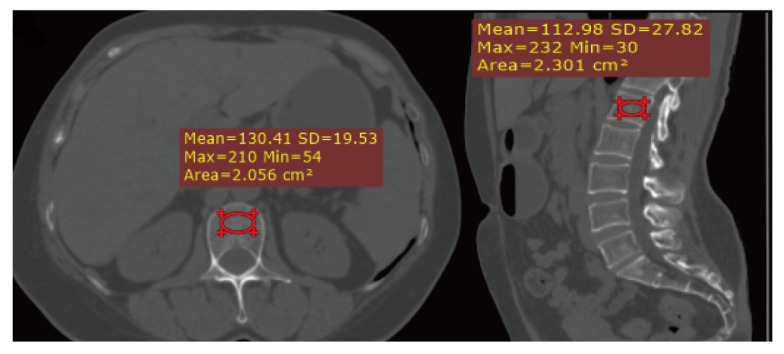
Coronal (**left**) and sagittal (**right**) CT images of lumbar spine (L1) in a 52-year-old women patient with pancreatic neuroendocrine tumors (PNETs).

**Figure 2 jcm-11-06701-f002:**
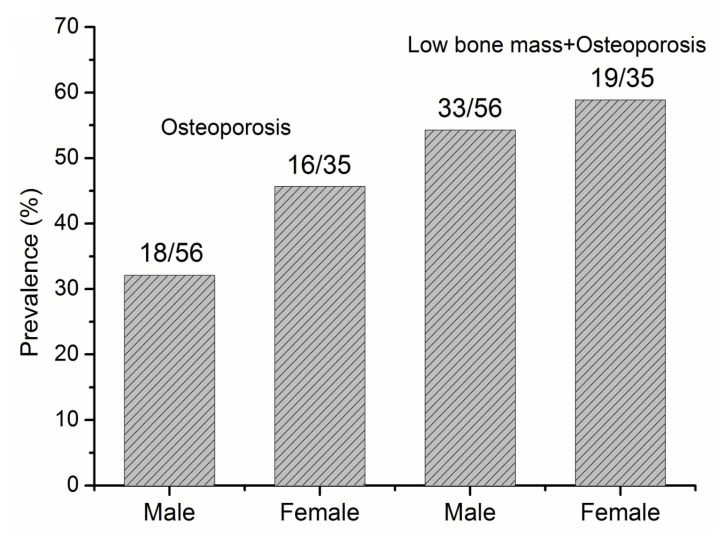
The prevalence of osteoporosis and bone loss (low bone mass + osteoporosis) in patients with pancreatic neuroendocrine tumors (PNETs).

**Figure 3 jcm-11-06701-f003:**
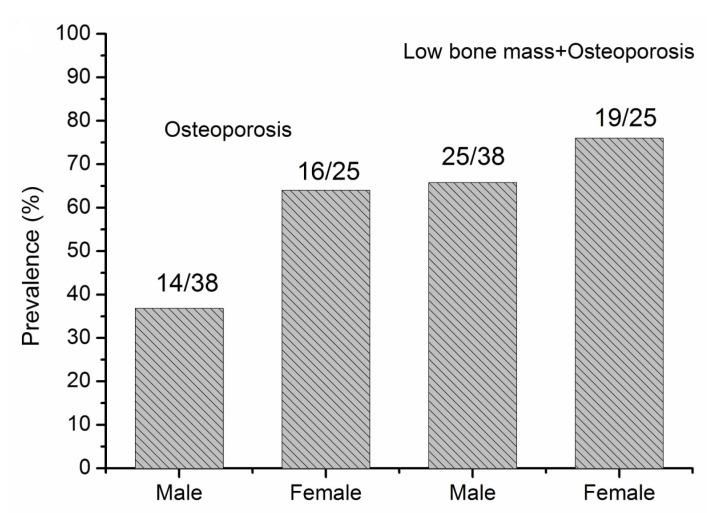
The prevalence of osteoporosis and bone loss (low bone mass + osteoporosis) in old patients (>50 years) with pancreatic neuroendocrine tumors (PNETs).

**Table 1 jcm-11-06701-t001:** Characteristics of patients.

	PNET (*n* = 91)
Age	55.76 ± 12.97
Gender (women/men)	35/56
Functional tumor	8
Clinical symptoms	
Abdominal pain	29
Weight loss	4
Jaundice	3
Back pain	4
Abdominal mass	0
Glucopenia	6
Asymptomatic	48
Others	8
Post-menopausal status *	27
TNM stage	
T1	27
T2	52
T3	12
N0	76
N1	15
CT value (HU)	158.0 ± 50.11
Location (head-neck/body-tail)	49/42
Size	3.14 ± 1.67
Grade (G1/G2/G3)	20/24/14
CT values (HU)	
>160	36
135–160	21
<135	34

CT: computed tomography; HU: Housfield units; PNETs: pancreatic neuroendocrine tumors (PNETs). * for women.

**Table 2 jcm-11-06701-t002:** Associations between variables and osteoporosis and low bone mass in pancreatic neuroendocrine tumors (PNETs).

	Osteoporosis		Low Bone Mass	
	Univariate (OR, 95% CI)	Multivariate (OR, 95% CI)	Univariate (OR, 95% CI)	Multivariate (OR, 95% CI)
Age	1.11 (1.05–1.17)	1.14 (1.03–1.20)	1.12 (1.06–1.18)	1.13 (1.04–1.22)
Gender (women vs. men)	1.78 (0.75–4.24)	2.36 (0.68–9.78)	0.83 (0.35–1.94)	0.81 (0.32–2.75)
Location	1.60 (0.53–4.85)	1.74 (0.49–7.69)	1.0 (0.43–2.00)	1.24 (0.43–4.36)
Diabetes mellitus	10.64 (1.32–118.7)	13.56 (1.02–132.4)	/	/
Grade (G3 vs. G1/G2)	0.45 (0.12–1.84)	0.36 (0.04–1.13)	1.28 (0.42–4.54)	0.54 (0.13–4.56)

CI: confidence interval; OR: odds ratio.

## Data Availability

All data generated or analyzed during this study are included in this published article.

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
