# Peer review of "Bone Loss in Patients with Pancreatic Neuroendocrine Tumors"

_jcm, 2022, doi:10.3390/jcm11226701_

Round 1
Reviewer 1 Report (Previous Reviewer 3)
Dear Authors,
this is a retrospective analysis on bone density deficit in patients with pancreatic neuroendocrine tumors and solid pseudopapillary tumors of the pancreas. According to the previous review, I've appreciate the details the authors have reported. However, the reason for an analysis of NET with exocrine disease of the pancreas have been analysed together is still unclear. Moreover, even if you have detailed the systemic treatments the exclusion criteria, it is still unclear if the patients underwent radical surgery or not. In this case, diabetes and the other factors of predisposition to osteoporosis are not necessary related to cancers (that seems your thesis) but to the treatments, because the patients, in other words, are healed.
I think the authors must to review the objective of their analysis.
Lastly, CT is not the standard to the bone density assessment.
Author Response
Dear Editor and reviewers,
Thank you very much for the critical review and the chance for revision. We have responded to the reviewer’s comments point by point and summarized the changes below. We have revised the manuscript accordingly and marked the changes with red color fonts. Based on the reviewer comments, we have deleted the related content for solid pseudopapillary tumors in the revised manuscript. Thus, the title was also changed to “Bone loss in patients with pancreatic neuroendocrine tumors”. We hope the revised manuscript has been improved to the quality that is suitable for publication now. We are looking forward to your favorable decision.
Reviewer Comments:
I reviewed the edited manuscript of the authors but I still have several perplexities.
First of all, even if they have added the exclusion criteria for the selection of the patients ("Those patients with distal metastasis, without CT examinations, and 61 who received any treatment, such as radiotherapy, chemotherapy or somatostatin before 62 CT examination were excluded from this study.") the authors have not understand what the problem is. They did not specify if included patients underwent radical surgery or not.
In the first case, the fact the patients had neuroendocrine or exocrine tumors (that I still don't understand why they analyzed together) is meaningless because the patients are, in other words, healed at the time of the CT scan for the assessment for bone density.
Thank you for your comments. We are sorry for the unclear description. Our patients did not receive any treatment, including radical surgery.
In the second, why they still analyze neuroendocrine and exocrine tumors together? It is well-known that these diseases are too much different.
Thank you for your comments. Because the bone loss in solid pseudopapillary tumors (SPTs) of the pancreas has not been reported. Therefore, we also showed the bone loss in SPT patients. Considered the reviewer comments, we deleted the solid pseudopapillary tumors of the pancreas in the revised manuscript.
In the discussion, indeed, the diabetes, surgery and other endocrine factors seemed to be associated to the bone density, regardless of the disease, so why focusing on these two rare diseases?
Thank you for your comments. The increase of incidence of PNET has increased during the past decades. Neuroendocrine tumors (NET) may have hormone hypersecretion which may affect bone metabolism. Bone health is usually neglected in patients with tumors. However, it is also valuable to know the bone health in tumor patients for patients management, especially considering the long-term survival of these patients. American Society of Clinical Oncology (ASCO) also suggests that cancer patients should receive a personalize bone mineral density test or taking pharmacologic interventions if necessary. Our research field main focused on pancreatic tumors. The bone loss in patients pancreatic cancer has been reported. Few reports were observed in PENT patients. Thus, we aimed to observed the bone loss in PNET patients. Although PNETs are rare diseases, performing such a study is acceptable. Please consider our explanations.
Our study provided several information for bone health in PNET patients Firstly, our results showed that prevalence of low bone mass or osteoporosis is high. Second, our results showed that PNET grade is not associated with bone loss, and age and diabetes play more important role.
Lastly, the use of CT as bone density assessment is not the standard.
Thank you for your comments. It would be better to evaluate the bone mass by DXA or QCT. However, QCT is not routinely performed. Several studies have showed that bone CT attenuation is high-positively correlated to BMD. And bone CT attenuation is also used to define bone mass(Pickhardt et al., 2013). Therefore bone CT attenuation is acceptable for bone mass evaluation. We have listed this as a limitation and talked about this issue.
Pickhardt P, Pooler B, Lauder T, del Rio A, Bruce R, Binkley N. Opportunistic screening for osteoporosis using abdominal computed tomography scans obtained for other indications. Ann Intern Med. 2013; 158(8):588-595.

Reviewer 2 Report (Previous Reviewer 2)
The paper has been improved, it is suitable for publication.
Author Response
Thank you.
Reviewer 3 Report (Previous Reviewer 1)
I think the manuscript can be published after the author's revision.
Author Response
Thank you.
This manuscript is a resubmission of an earlier submission. The following is a list of the peer review reports and author responses from that submission.
Round 1
Reviewer 1 Report
The manuscript is an attempt to evaluate the prevalence of low bone density in patients with pNETs or pancreatic preudopapillary tumors and identify relevant risk factors. However, there are a some factors such the previous pancreatectomy, the funcionality of the tumors or the disease duration and status (localized/metastatic) that could affect bone mass and are not reported.
In Figure 2A, the number of the patients are somehow wrong.
Reviewer 2 Report
The paper titled “Bone loss in patients with pancreatic neuroendocrine tumors and solid pseudopapillary tumors of the pancreas” from He Tong et al. evaluated the prevalence and the potential risk factors of bone loss and osteoporosis in two different cohort of patients, one with pancreatic neuroendocrine tumors (PNETs, n= 91) and one with solid pseudopapillary tumors of the pancreas (SPTs, n= 75).
The paper focused on interesting and original topic, yet unexplored in the literature but with a relevant clinical impact due to the increase of incidence of PNET and the long-term survival of these patients. Anyway, in the evaluation of this work some major concerns emerged:
Major concerns:
1. Introduction: authors should extend the background considering these two points:
1. Diabetes and obesity are potential risk factors for PNET occurrence (Feola T et al. doi.org/10.1007/s40618-021-01715-0) and they also may have an impact on bone health (Leslie WD PMID 28917003, Bai J et al. PMID 31768878; Koromani et al PMID 31658976); moreover diabetes is associated to a more advanced and progressive disease (Feola T et al. doi.org/10.1007/s40618-021-01715-0; Muscogiuri et al. doi.org/10.1007/s12020-020-02331-3).
2. PNET may metastasize in bone, increasing the risk of break or fracture (Altieri et al. 10.3390/cancers11091332).
2. Methods: authors should specify that the data collection was retrospective. I suggest extending the description of inclusion and exclusion criteria, and also the statistical methods for the identification of risk factors (OR calculation)
3. Results: patients’ characteristics are poorly described. PNET are localized or metastatic? Please add TNM stage. How many patients have a functioning PNET? Please add the specific syndrome. Moreover, I think that could be useful specify how many women are in post-menopausal state and if in men hypogonadism (overt or subclinical), was excluded because low testosterone levels, may have a great impact of bone and body composition (Kazuyoshi Shigehara et al. 10.3390/jcm10030530; Isidori et al 10.1111/j.1365-2265.2005.02339.x; Giannetta et al. doi:10.1016/j.beem.2011.12.005). Data about fractures, bone turnover markers, vitamin D status, medical therapies, and other risk factors such as smoking or corticosteroid use, are also lacking. Therefore, if these data are not available the authors should add a comment in discussion and including that in the limitations.
4. Discussion: I suggest considering to add more comments on the relationship between diabetes and PNET and on the link between diabetes and bone loss, considering also the disease stage. Moreover, considering that age is a risk factor for bone loss I suggest commenting the role of gonadal status both in women and in men. The authors should stress study limitations, suggesting also the need of prospective studies on this topic (see Results)
Reviewer 3 Report
This is a well written retrospective analysis of a multifactorial event that may occur in patients with neuroendocrine neoplasms and other pancreatic cancers. Despite the analysis could be interesting, in my opininon, the study design doesn't reach the quality to give some important clinical tools to the reader.
The folowings are my observations:
54-61: were PNET patients naive from any treatment? Or they could have already receive some TKI or other treatments?
Period of collection patients is lacking.
It is not clear why the authors analyzed together NEN and SPTs. I suggest the authors to remove the sample of SPTs, also according to the special
63-72 It is not clear if the CT scan has been obtained at the time of diagnosis or after. It is different if the patients had osteopenia/osteoporosis at the diagnosis or after and it is not clear if they had a worsening of the problem over the disease journey.
Table 1: when you define G1, G2 and G3 what do you mean with “G3”? Well differentiated G3 or NENs G3 (well and poorly differentiated)? Which NENs classification do you refer to?
163-164: The sentence seems to be redundant and not related to the biology of the disease (the patient who undergoes a radical surgery is not a patients with this neoplastic disease anymore).
Which news/clinical tools offer the results of this study?